# Efficient Photodegradation of Dyes from Single and Binary Aqueous Solutions Using Copper(II) Coordination Polymers

**DOI:** 10.3390/molecules30081652

**Published:** 2025-04-08

**Authors:** Ildiko Buta, Maria Andreea Nistor, Simona Gabriela Muntean

**Affiliations:** “Coriolan Drăgulescu” Institute of Chemistry, Romanian Academy, 24 Mihai Viteazu Bvd., 300223 Timisoara, Romania; ildiko_buta@acad-icht.tm.edu.ro (I.B.); nistor_andreea1990@yahoo.com (M.A.N.)

**Keywords:** dyes, photocatalytic degradation, coordination polymer, binary solution, kinetic studies

## Abstract

The present study reports the application of three copper(II) coordination polymers, namely ^1^_∞_[Cu_3_L_2_(N_3_)] CH_3_COO (**CP1**), ^1^_∞_[Cu_3_L_2_(NO_3_)]NO_3_·2CH_3_OH·2H_2_O (**CP2**), and ^1^_∞_[Cu_3_L_2_(H_2_O)](ClO_4_)_2_ (**CP3**), where **H_2_L** stands for N,N′-bis[(2-hydroxybenzilideneamino)propyl]-piperazine) as catalysts for photocatalytic degradation of Acid Orange 7 and Methyl Orange dyes from single and binary aqueous solutions. The influence of the photocatalyst nature, hydrogen peroxide presence, reaction time, dye concentration, and catalyst dose on the photodegradation efficiency was studied. Under visible light irradiation, complex **CP1** demonstrated the highest photodegradation efficiency of 92.40% and 80.50% towards Acid Orange 7 and Methyl Orange, respectively. The kinetic studies indicated that the photodegradation process followed a pseudo-first-order kinetics. The highest rate of the degradation process was obtained when **CP1** is used, and the necessary time for the degradation of the dyes increases with increasing concentration of the dye solutions. The degradation efficiency of more than 75% after five recycling/reuse cycles of **CP1** and the yields higher than 72% obtained for the degradation of dyes from the binary system demonstrate the photocatalytic capacity of **CP1**. A photocatalytic oxidation mechanism was proposed and the stability of the **CP1** complex before and after the photodegradation process of dyes, both from simple and binary solutions, was investigated and confirmed.

## 1. Introduction

The rapid expansion of cities, consumerism, and the development of the textile industry in recent decades has caused serious environmental pollution. One of the most serious problems has become water pollution, which causes long-term damage to the environment [1]. Dyes are organic compounds widely used especially in the textile and food industries. The discharge of colored textile wastewater into collecting canals or even into rivers and streams has extended and become the main source of pollution worldwide [2]. The dye molecules do not degrade as colored wastewater is discharged into rivers and then into seas and oceans [3,4,5]. This causes the gradual accumulation of dyes in the water and inhibits sunlight from penetrating the surface of water. This fact affects the ability of plants to carry out photosynthesis, determining the reduction of the oxygen content in the water and thus causing the death of aquatic flora and fauna. For all these reasons, the treatment of colored wastewater is of great importance and interest both for water purification and for its reuse [6]. In the last decades, various methods of treating water contaminated with azo dyes have been investigated, including physical, chemical, biological techniques, and others [6,7,8].

Photodegradation of water pollutants represents advanced oxidation processes that include ultraviolet or visible light irradiation, successfully applied for the degradation of chemical substances; e.g., dyes from industrial effluents [9,10,11]. In the last years, the design and synthesis of new materials with good photocatalytic properties and visible light responsiveness have been highly demanded and developed. To address this challenging issue, many studies were carried out to improve the properties of existing photocatalytic materials or to develop new potential materials such as metal oxide nanocomposites [12], multi-component oxides [13] polymeric nanostructures [14], modified zeolitic imidazolate frameworks nanoparticles [15], geopolymers [16], etc. Colored industrial wastewater usually contains mixtures of dyes. In the last decade, studies have focused on the removal of dyes from multicomponent solutions by adsorption [17,18,19]. Based on data from the literature, we found that the potential application of a catalyst in a multipollutant system has been studied in few cases. Recently, different materials have been applied for the photocatalytic degradation of dyes from binary systems with some interesting results. For instance, Zhang and coworkers [20] reported the fabrication of iron-doped manganese oxide nanoparticles for the degradation of Indigo Carmine and Rhodamine B from a binary mixture under solar light irradiation. Ayodhya [21] reported the degradation of Fluorescein/Rhodamine B from a binary dyes system, in the presence of Ag-CuO composites, under visible light irradiation. Verma and coworkers [22] investigated the kinetics of photocatalytic degradation of single and binary dye mixtures regarding concentration, pH, and extent mineralization to identify the best working conditions for simultaneous degradation. Usman’s group reported selective photocatalytic degradation of Methylene Blue over Rhodamine B in the presence of an anatase TiO_2_ catalyst, under UV light irradiation [23]. However, to the best of our knowledge, no study on the photocatalytic degradation of dyes in multicomponent systems using coordination polymers is reported so far.

Coordination polymers (CP) are a class of compounds with extended structures obtained from the self-assembly of metal ions and organic bridging ligands [24,25,26]. CPs has aroused great interest due to their properties suitable for potential applications in fields like magnetism [27], luminescence [28], gas adsorption [29], chemical sensors [30], and lately for the promising results in sustainable polluted wastewater treatment [31,32,33,34]. These properties are strongly related to the topology of the CP’s structures that can be determined by an appropriate choice of the metal ion, the preorganization capacity of the ligand, and the nature of the counterions [25,26]. Several desirable properties, including semiconducting nature [35], visible light responsiveness [36], and tunable surface area [37], make CPs a promising photocatalyst. In the literature, there are several papers reporting the heterogeneous photodegradation of dyes using copper(II)-based coordination polymers [38,39,40], and by applying UV-Vis spectrometry as a viable method for studying photocatalysis over time [41,42]. Therefore, as a continuation of our work [43], in the present study, the photocatalytic properties of three Cu^II^ coordination polymers, namely ^1^_∞_[Cu_3_L_2_(N_3_)]CH_3_COO (**CP1**), ^1^_∞_[Cu_3_L_2_(NO_3_)]NO_3_·2CH_3_OH·2H_2_O (**CP2**), and ^1^_∞_[Cu_3_L_2_(H_2_O)](ClO_4_)_2_ (**CP3)**, where **H_2_L** stands for N,N′-bis[(2-hydroxybenzilideneamino)propyl]-piperazine), have been reported. In the present study, the degradation capacity of **CP1**–**CP3** was evaluated for the removal of two azo dyes from single and binary systems. Moreover, the factors that influence the degradation process were analyzed and the kinetic parameters of the process were determined.

## 2. Results and Discussion

### 2.1. Characterization of **CP1**–**CP3**

The detailed information about the synthesis and structural characterization of **CP1–CP3** are presented in the previous paper [43]. Briefly, the **CP1**–**CP3** complexes were obtained by reacting appropriate copper(II) salts with **H_2_L** in a 2:1 stoichiometric ratio, according to Figure 1. The crystallographic investigation revealed that **CP1**–**CP3** consist of trinuclear complex entities, {Cu_3_L_2_}^2+^, connected via azido (**CP1**), nitrato (**CP2**), and phenoxido (**CP3**) bridges.

#### 2.1.1. Thermal Properties of **H_2_L** and **CP2**

Due to the fact that the perchlorate and azide metal complexes are potentially explosive, the thermal properties were investigated only for **H_2_L** ligand and **CP2** [44]. Thermogravimetric analysis measurements performed on both **H_2_L** ligand and **CP2** in nitrogen/air conditions shows a relatively good thermal stability, with no decomposition processes below 200 °C (Figure 2). The decomposition of the **H_2_L** occurs in three stages. The first stage is found in the range of 25–350 °C, with the mass loss of 48.64% (calcd. 48.04%) corresponding to piperazine, two propyl chains and two nitrogen atoms. The second and third stages are observed in the range of 350–500 °C and 500–1000 °C and are associated with the weight losses of 29.79% (calcd. 29.42%) and 7.29% (calcd. 7.83%), respectively corresponding to the loss of aromatic fragments. For complete oxidation of the organic residues, the system was supplied with air and a final mass loss of 14.28% (calcd. 14.71%) was observed.

The thermogram of **CP2** shows two major stages of decomposition and a dehydration process in the range of 30–80 °C of 2.93% (calcd. 3.02%) corresponding to the loss of two water molecules present in the outside coordination sphere. The first stage of 42.94% starts at 230 °C and continues up to 430 °C and the second one of 37.06% starts after 450 °C and completes at 800 °C. By supplying the system with air, complete oxidation of organic residues occurs and the resulting product coincides for three copper and three oxygen atoms (found 20.00%; calc. 19.55%).

#### 2.1.2. Band Gap Analysis

The band gap of a semiconductor photocatalyst is an important parameter for determining its degradation capacity and refers to the energy difference between the highest valence band and the lowest conduction band [45]. In general, a decrease in the band gap increases the photocatalytic activity.

The band gaps for **CP1**–**CP3** were calculated from UV-Vis absorption spectrum, using Tauc plots and Tauc’s Equation (1) [33]:(1)$${\left(\alpha h\upsilon \right)}^{n}=k\left(h\upsilon -E_{g}\right)$$
where *α* is the optical absorption coefficient (2.303A/*t*); A—absorbance; *t*—thickness of the material; *hν*—1240/wavelength; *E_g_*—bandgap energy; and *n* is 2 for direct transition.

Representing the dependence of (α*h*ν)^2^ as a function of the photon energy (*h*ν) and extrapolating the linear portion of the curves to the zero absorption coefficient value, the values of the energy bands for the investigated materials were obtained. The *E_g_* values were 3.09 eV for **CP1**, 3.15 eV for **CP2**, and 3.12 eV for **CP3**, comparable with other CPs, indicating that the CPs might be used as catalysts. The *E_g_* value obtained for **H_2_L** ligand was 3.68 eV.

### 2.2. Photocatalytic Studies

The catalytic activities of the synthesized complexes (**CP1**–**CP3**), toward photodegradation of industrial dyes were explored. The influence of different process parameters such as nature and quantity of the catalyst and initial dye concentration on photodegradation efficiency were investigated.

#### 2.2.1. Influence of the Nature of the Photocatalyst

To study the photodegradation of AO7 and MO dyes, we first investigated the influence of the nature of the material applied as a photocatalyst. Coordination polymers **CP1**–**CP3** (1 g/L) were introduced into aqueous dye solutions (30 mg/L), photodegradation studies being performed at room temperature, solution pH, under visible light irradiation, for 6 h. In order to evaluate the photocatalytic capacities of the investigated coordination polymers, two additional control studies were performed working under the same conditions (irradiation with visible light, pH of the solution, room temperature); first, in the absence of CPs and second in the presence of the ligand (**H_2_L**). The obtained results are presented in Figure 3.

The results (Figure 3) showed that the degradation of AO7 and MO solutions under visible light irradiation in the absence of a photocatalyst (CPs) was not detected (R < 2%).

The ligand (**H_2_L**) exhibited 10.42 and 9.82% removal of AO7 and MO in 200 min, respectively, indicating that the catalyst’s properties of **H_2_L** were insufficient for the degradation of investigated dyes. At the same time, a significant degradation of the investigated dyes occurred in the simultaneous presence of **CP1**–**CP3** and visible light.

After 150 min under irradiation at 546 nm, the degradation efficiency of AO7 dye was higher than that for MO degradation (Figure 3a). The degradation efficiencies of AO7 reached 92.40% for **CP1,** 71.80% for **CP2**, and 89. 30% for **CP3**. For the case of MO, the obtained values were 80.50% for **CP1**, 47.00% for **CP2**, and 57.45% for **CP3**. As can be seen the catalytic efficiency increased in the order **CP2** < **CP3** < **CP1**, consistent with the values obtained for the band gap energy (E_g_), indicating that **CP1** had the highest photocatalytic activity.

The use of H_2_O_2_ for the degradation of colored pollutants is useful as an auxiliary oxidant, due to the formation of additional highly reactive hydroxyl radicals [46]. Thus, 30 µL hydrogen peroxide 30% was added to the system as an oxidizing agent and the experiments were performed under the same conditions (pH, temperature, quantities). The obtained results under visible light irradiation with (Figure 4b) and without (Figure 4a) H_2_O_2_ highlights the fact that the removal efficiency of AO7 and MO increased slightly in the presence of hydrogen peroxide.

The removal efficiency of AO7 increased with the addition of H_2_O_2_ from 92.40% to 93.81% for **CP1**, from 71.80% to 76.60% for **CP2**, and from 89.03% to 91.10% for **CP3** (Figure 4b). For the case of MO, the presence of hydrogen peroxide generated an increase in the degradation efficiency from 80.50% to 84.82%, from 47.00% to 51.70% and from 57.45% to 71.43% in the case of using **CP1**, **CP2**, and **CP3**, respectively. The enhancement of photodegradation efficiency may be attributed to the presence of a larger quantity of hydroxyl radicals from the photolysis of H_2_O_2_ that can react with the dye molecules.

As seen in Table 1, compared to the dye photodegradation activity of other recently reported coordination polymers (Table 1), the investigated complexes show similar or even higher photodegradation efficiency.

Overall, taking into account that the addition of hydrogen peroxide into the system slightly increased dye degradation by 6.70% for AO7 and 24.30% for MO along with the additional costs and possible pollution involved [51], we considered that it was more feasible to continue further studies in the absence of H_2_O_2_.

#### 2.2.2. Evolution in Time of the Photodegradation Process

The degradation process in time was followed by measuring the intensity of the characteristic absorption band at 483 nm for AO7 and at 463 nm for MO, respectively. The changes over time, in the absorption spectra of the AO7 and MO dyes during the photocatalytic degradation process, under visible light irradiation using **CPs** as catalysts, are presented in Appendix A. In the presence of the complexes, the intensity of the absorption peak of AO7 and MO decreased significantly in 180 min revealing that the photocatalytic reaction progressed. The change of color over time is also exemplified in Appendix A. The clear solutions obtained in the case of using CP1 should be noted, this can be attributed to the greater photocatalytic degradation capacity of this compound.

As shown in Figure 5, during the first 30 min, adsorption occurs on the catalyst surface. Previous studies related to the dyes photodegradation have shown that a time of 30 min is suitable to reach equilibrium in the adsorption process [52]. After the photocatalytic reaction, in the presence of CPs, the concentration of dye solutions decreases significantly in 100–150 min, indicating the efficiency of photodegradation of AO7 and MO dyes from aqueous solutions.

In the presence of **CP1**–**CP3**, the intensity of the absorption peak of AO7 and MO decreased significantly in the time interval of 100–150 min revealing that the photocatalytic reaction progressed. As the photocatalytic degradation occurs, the concentration of the dye solutions slowly decreases, indicating a possible surface coverage of the catalyst in time which can hinder the exposure to light irradiation and the formation of radicals for further oxidation of the pollutants [53]. Control experiments (without CP or in the presence of ligand) showed no degradation of the dyes over time.

#### 2.2.3. Effect of the Catalyst Dosage

Since the best results were obtained using **CP1** as a catalyst, further studies were carried out using this compound. In order to highlight the efficiency of **CP1** as a catalyst, the influence of the catalyst dosage on the dyes’ degradation efficiency was further investigated. For this study, amounts of 0.5, 1, 2, and 3 g/L were used, keeping all other operating conditions constant. Figure 6a illustrates the degradation efficiency of AO7 and MO dyes in the presence of different amounts of coordination polymer **CP1**. As was expected, the dye degradation efficiency increased with increasing **CP1** dosage. The efficiency increased from 70.54% to 92.40% for AO7 and from 54.54% to 80.50% for MO, with the increase of **CP1** dosage from 0.5 to 1 g/L, possibly due to the increase in the number of active sites on the catalyst surface available for dye molecules.

The use of amounts greater than 1 g/L of **CP1** causes an insignificant increase in the degradation yield: ~2% for AO7 and 4.5% for MO, respectively. It can be assumed that an excess of **CP1** beyond the optimum amount may induce a shading effect by lowering the light transparency of the solution and reducing the amount of sunlight that reaches the active sites of the catalyst [54]. Therefore, the optimal dose of **CP1** catalyst selected for further studies was 1 g/L.

#### 2.2.4. Effect of Initial Concentration of Dye Solutions: Kinetic Studies

Photodegradation is known to be influenced by the concentration of organic dye in a solution, which is closely related to dye aggregation [55]. As the dye concentration increases, the dye molecules form aggregates. The higher the concentration of the dye solution, the larger the size of the dye aggregates [56,57]. Aggregates are adsorbed on the substrate surface making it difficult to access the catalyst surface, leading to longer photodegradation times and yields.

Different concentrations were selected: 15, 30, 45, and 60 mg/L to investigate the influence of the initial dye concentration on the photodegradation efficiency, maintaining constants of temperature and solution pH. Figure 6b,c illustrate the photocatalytic degradation of the investigated dyes for different values of the initial concentration. As can be seen in Figure 6b,c and Table 2, the degradation efficiency increased slightly with the increase in the dye concentration from 15 to 30 mg/L but decreased significantly for a further increase in the concentration of the dye solutions.

This behavior is in agreement with reported studies for different types of catalysts and organic dyes [55] and indicates that, for a certain amount of catalyst, with increasing dye concentration the number of active sites available for photocatalysis decreases [58,59]. As the concentration increases, more dye molecules are adsorbed on the catalyst surface. Moreover, the large number of the adsorbed dye molecules hinders the absorption of visible light and, consequently, prevents the initiation of the photo-oxidation process [55].

Degradation yields of 92.40% (AO7) and 80.50% (MO) were obtained working under optimal conditions for 30 mg/L dyes solutions, this being considered as the optimal concentration for both investigated dyes. Additionally, the obtained results highlighted the fact that solutions with low dye concentrations (15 mg/L) can be discolored faster than the most concentrated ones (60 mg/L), the time required for degradation increasing with the increase in the initial concentration of the dye solutions (Table 2).

Kinetic studies on the photocatalytic degradation of AO7 and MO dyes over time were carried out using 10 mg of CPs catalyst in 10 mL of dye solutions. Kinetic studies were also performed using **CP1** as a catalyst, for different dye concentrations of 15, 30, 45, and 60 mg/L. The kinetic of dye photodegradation was investigated using the Langmuir–Hinshelwood model, Equation (2) [60].(2)$$ln\left(\frac{C}{C_{0}}\right)=-k_{app}t$$
where *C*_0_ is the initial dye concentration (mg/L), *C* is dye concentration at time *t* (mg/L), *t* is the irradiation time (min), and *k_app_* is the first-order rate constant (min^−1^).

From the graphical representation of the experimental data as ln(*C*/*C*_0_) versus time (Appendix A), the rate constant (*k*) was determined by linear fitting based on regression analysis, and the obtained data are presented in Table 3. For each dye and each compound included in this investigation, the obtained R^2^ and SD values are listed.

The obtained results showed that the degradation process of AO7 and MO using **CP1**, **CP2**, and **CP3** followed a pseudo-first-order kinetic. The rate constants obtained indicated that the speed of degradation process increases in the order **CP2** < **CP3** < **CP1**.

The theoretical data obtained for the first-order rate constants (*k*) (Table 3) show that, with the increase in dye concentration, the necessary time for the photodegradation of investigated dyes increases, which confirms the results obtained experimentally (Table 2). With the increase in dye concentration, the values obtained for *k* decrease, indicating lower degradation process rates.

#### 2.2.5. Recycling of Photocatalyst

To evaluate the stability of the **CP1** catalyst during the dye degradation process and its reuse, five photocatalytic degradation cycles of AO7 and MO were performed. After each cycle, the catalyst was washed with ethanol, dried, and then reused.

As the number of cycles increases, the efficiency of the photocatalytic degradation decreases, due to the occupation of the active centers on the catalyst surface by the adsorption of dye molecules (Figure 7a). Qualitative estimation of dye degradation can be emphasized by the decoloration of the dye solutions highlighted in Appendix A.

From the first to the third reutilization, a reduction in the photodegradation efficiency of 18.70% in the case of AO7 and 13.60% for MO (Figure 7a) was observed. From the third to the fifth cycle, the loss in efficiency was 13.95% for AO7 and 16.30% for MO, respectively. After five simultaneous adsorption–photocatalytic cycles, the capacity of **CP1** was still considerable. The decrease in degradation yield after five reuses was about 30% for AO7 and 27% for MO, and the average degradation efficiency for the five cycles was 76.76% for AO7 and 67.53% for MO, respectively, demonstrating the reusability of **CP1**.

#### 2.2.6. Photodegradation of Binary Dye Mixture

To highlight the efficiency of a catalyst, it is important to examine its performance in the decomposition of dyes in mixtures, which simulate a real textile effluent [46,61].

Based on our knowledge and the literature survey carried out, only a few researchers have investigated the potential of a dye degradation catalyst in a multicomponent system. Moreover, until now, there is no mention in the specialized literature related to the application of coordination polymers for the degradation of dyes from binary systems. Therefore, for future potential industrial applications, in addition to the regeneration and reuse studies of **CP1**–**CP3**, the degradation studies of dyes from multicomponent mixtures are very important. For these studies, the photocatalytic degradation of AO7 and MO dyes from binary solutions was evaluated, working under visible light irradiation, at a speed of 300 rpm, 25 °C, solution pH, using an amount of 10 mg of **CP1**–**CP3**, for a volume of 10 mL dye solution. The experiments were performed for a dye concentration of 30 mg/L, in a volume ratio of 1:1 with, and without the addition of 30 µL H_2_O_2_.

UV-Vis spectrophotometry was used to follow the decrease in absorbance over time (Appendix A), and implicitly the discoloration of the dye solutions due to the degradation of the dyes. After 240 min, the absorbance decreased due to discoloration of the dye solution under visible light, using **CP1**.

The degradation of the mixture of binary dyes was evaluated based on the degradation efficiency (Equation (1)) calculated at the value of the maximum wavelength of the binary solution (474 nm) as well as at the wavelength of each dye: AO7 (483 nm) and MO (463 nm). For the binary solution, lower degradation efficiencies were obtained compared to the single solutions (Figure 7b, Table 4) for all tested compounds (**CP1**–**CP3**). This is due to the fact that the two azo dyes in the binary solution compete for the active centers on the **CP1**–**CP3** surface, available for adsorption and degradation.

The degradation of the dyes in the binary system followed the same trends as in the single solutions. The highest degradation efficiency of 73.83% and 83.54% (H_2_O_2_ added), respectively, was obtained using **CP1**. It is notable that, also in the case of using **CP3** as a catalyst, degradation efficiencies higher than 53% (H_2_O_2_ added) were obtained for the mixture of dyes from the binary system.

If we discuss from the point of view of the individual dyes in the binary system, yields higher than 72% were obtained for both dyes using **CP1**. When H_2_O_2_ was added, similar to single dye solutions, the degradation efficiency of both dyes in the binary system increased to over 81% using **CP1** and over 50% using **CP3**, respectively (Figure 6b).

Overall, **CP1** showed the highest photocatalytic degradation of both single dyes and the mixture of dyes in the binary system, compared to **CP2** and **CP3**.

The obtained results are promising and encouraging, indicating that **CP1** is a viable catalyst for subsequent application on colored industrial wastewater.

### 2.3. Material Stability

The stability of a photocatalyst is crucial for the practical applications in environment remediation. Therefore, in order to estimate the stability of the studied material, the IR spectra of **CP1** before and after photocatalytic reactions were recorded (Figure 8).

The IR spectrum of the solid residue of **CP1** isolated from the reaction mixture via centrifugation and dried in air exhibits a very strong absorption band situated at 2034 cm^−1^ characteristic to the N_3_^−^ anion and a sharp signal at 1624 cm^−1^ assigned to the ν(C=N) stretching vibrations. The presence of vibrational bands at 1539 and 1401 cm^−1^ is typical of antisymmetric and symmetric ν(COO^−^) stretching frequencies, respectively, while the difference ∆ν of 138 indicates the uncoordinated mode of the carboxylate group [62]. The band at 1328 cm^−1^ is assigned to the ν_aryl-O_ vibration. The medium band observed at 759 cm^−1^ corresponds to the C-H out-of-plane bending vibration of the substituted phenyl ring. The IR spectra of the **CP1** after photodegradation of AO7 and MO from a single and binary system display the same pattern with the original one, demonstrating the stability of the complex in the photocatalytic process.

The confirmation of **CP1’s** stability along with the good results from the presented recycle studies (see Section 2.2.5) recommend **CP1** as an efficient catalyst for the degradation of dyes from aqueous solutions.

### 2.4. Photocatalytic Mechanism

The possible mechanism for the photocatalytic process is based on the semiconductor’s capacity to generate electron–hole pairs able to promote oxidation and reduction reactions followed by organic dyes degradation [63]. Electron–hole pairs are generated upon visible light irradiation; thus, (*i*) electrons from the valence band are excited to the conduction band leaving holes (*h^+^*) with oxidizing character capable of generating hydroxyl radicals (•OH); (*ii*) in the conduction band, photoinduced electrons (e^−^) with reducing character are formed, capable of generating superoxide radical anions (^•^O_2_^−^) (Figure 9) [64,65,66].

In order to confirm the assumed radical-mediated path, a trapping experiment with disodium ethylenediaminetetraacetate (EDTA-2Na) as holes *h^+^* scavenger was performed. Upon addition of 10 mg EDTA in the photocatalytic reaction, the degradation rate of AO7 and MO in the presence of **CP1**–**CP3** decreased to 2–16%, suggesting only a very small quantity of *h^+^* in the system (Appendix A). There are two possibilities for photodegradation inhibition: directly, by blocking the attack of the holes on the pollutant and indirectly by blocking the formation of •OH radicals which oxidize the pollutant. However, the experiment shows that in this case, *h^+^* and •OH radicals are the main active species in the AO7 and MO photodegradation. Based on these results, a possible mechanism that follows the photocatalytic oxidation pathway can be proposed [67,68,69,70]. The reactive oxygen species generated could oxidize the AO7 and MO dyes. The possible degradation mechanism consists of several steps involving cleavage of the azo bond (–N=N–) generating β-naphthol, N,N-dimethyl-p-phenylenediamine, and sulfanilic acid [71,72], followed by further degradation of the intermediates to carbon dioxide (CO_2_), water (H_2_O), and other small organic molecules.

## 3. Materials and Methods

### 3.1. Synthesis of Coordination Polymers

Different solutions of **H_2_L** (1.22 mmol) in CHCl_3_/CH_3_OH (1:1 *v*/*v*, 60 mL) were reacted with Cu(NO_3_)_2_·3H_2_O (2.44 mmol) or Cu(ClO_4_)_2_∙6H_2_O (2.44 mmol) or Cu(CH_3_COO)_2_∙H_2_O (2.44 mmol) and NaN_3_ (2.44 mmol) dissolved in CH_3_OH (40 mL) and Et_3_N (3.7 mmol). Dark green single crystals suitable for X-ray analysis were obtained by slow evaporation of the mother liquor [43]. Elemental analysis calcd. for **CP1**: C_50_H_63_Cu_3_N_11_O_6_ (1104.73) C, 54.36; H, 5.75; N, 13.95%. Found: C, 54.72; H, 5.79; N, 13.76%; **CP2**: C_50_H_72_Cu_3_N_10_O_14_ (1227.79) calcd.: C, 48.91; H, 5.91; N, 11.41%. Found: C, 48.49 H, 5.78 N, 11.95%, **CP3**: C_48_H_62_Cl_2_Cu_3_N_8_O_13_ (1220.57): C, 47.23; H, 5.12; N, 9.18%. Found: C, 47.81, H, 5.23; N, 8.92%.

### 3.2. Materials and Physical Measurements

All reagents were used as purchased from commercial suppliers, without further purification. N,N′-bis[(2-hydroxybenzilideneamino)-propyl]-piperazine (**H_2_L**) was synthetized following the procedure described earlier [73]. Caution: Salts of perchlorate and azide and their metal complexes are potentially explosive and should be handled with great care and in small quantities.

In order to identify the synthesized samples, elemental analyses (C, H, N) were performed on an Elementar UNICUBE CHNS analyzer (Elementar Analysensysteme GmbH, Langenselbold, Germany), with helium as the carrier gas. To identify the functional groups of the material before and after photodegradation, Fourier-Transform Infrared (FTIR) spectra were recorded on a Cary 630 FTIR from Agilent Technologies (Santa Clara, CA, USA), using KBr pellets, in the range of 400 to 4000 cm^−1^. The TG data were obtained using a Discovery TGA 5500 analyzer from TA Instruments (New Castle, DE, USA) under a continuous flow of N_2_, heating rate of 10 °C/min, starting from 25 °C up to 1000 °C for **H_2_L** and from 25 °C up to 800 °C for **CP2**. After that, isothermal heating under an air flow for 10 min was maintained.

Two anionic azo dyes, Acid Ornge 7 (AO7; C_16_H_11_N_2_NaO_4_S, C.I. 15510) and Methyl Orange (MO; C_14_H_14_N_3_NaO_3_S), were selected as potential colored pollutants. Acid Orange 7 is widely used in the textile industry for dyeing wool, silk, and polyamide fibers in the leather and paper-processing industry. It can also be used both as an indicator and for biological dyeing. Methyl Orange is one of the most widely used pH indicators and titration indicators in the pharmaceutical industry and in analytical chemistry. The wide use of these dyes does not mean that they are not toxic or potentially hazardous compounds. Azo dyes contain aromatic groups (benzene rings) and azo groups in their molecules, which are toxic, carcinogenic, and teratogenic with a negative impact on the environment [74,75]. In addition, the presence of these dyes in wastewater discharged into sewers or rivers causes their coloring and therefore the deterioration of water quality [76]. The structures, the characteristics, and the toxicity of the selected dyes are presented in Table 5.

### 3.3. Photocatalytic Studies

The photocatalytic performances of the complexes **CP1**–**CP3** were investigated for the degradation of two organic dyes at room temperature, in a UV chamber with a 500 W Hg lamp providing 546 nm irradiation. Dye solutions of different concentrations, 15, 30, 45, 60 mg/L, were prepared by dilutions from a stock dye solution with a concentration of 200 mg/L. Experiments were performed at the natural pH of the dye solutions: 6.8 for AO7, and 6.4 for MO. The photocatalyst was dispersed in an aqueous solution of dye and stirred at 200 rpm in the dark for 30 min, to reach the adsorption–desorption equilibrium. Then, the mixture was irradiated using visible light under continuous magnetic stirring, at room temperature, in order to accomplish the dye photodegradation. At different time intervals, samples of solution were taken, and the catalyst particles were separated by centrifugation. Using a UV-vis spectrophotometer (ABLE-JASCO, Cluj, Romania), the absorbance at the maximum wavelength was measured: 483 nm for AO7, 463 nm for MO, and the concentration of the investigated dye in solution at different time periods was determined. Using the obtained data, the degradation efficiency was calculated with Equation (3):(3)$$R=\frac{\left(C_{0}-C\right)}{C_{0}}\cdot 100$$
where *C*_0_ and *C* represents the concentration of dye at the beginning, and after photodegradation (mg/L).

The control experiments were carried out under the same working conditions (natural solution pH, room temperature), but without the addition of the catalyst and also in the presence of the ligand.

The regeneration and reuse experiments of the investigated coordination polymers were carried out over five cycles under similar reaction conditions.

## 4. Conclusions

Three coordination polymers **CP1**–**CP3** were used in the photocatalytic process of AO7 and MO dyes from single and binary aqueous solutions. The obtained experimental data show that the photocatalytic activity increases in the following order: **CP2** < **CP3** < **CP1**, for both investigated dyes, consistent with the CP’s band gaps. In the larger context of dye removal efficiency studies, **CP1** stands out. This coordination polymer has an excellent ability to almost completely remove azo dyes from aqueous solution, the degradation efficiency being reduced by increasing the dye concentration and decreasing the amount of **CP1** used. Degradation efficiencies of 93.81% and 84.82% were obtained in the presence of H_2_O_2_, respectively, and over 90% for AO7 and 80% for MO in the absence of H_2_O_2_, indicating the high degradation capacity of **CP1** under simple and normal working conditions. The catalytic kinetics were described by a pseudo-second-order kinetic model, and the highest speed of degradation process was obtained for **CP1**.

An interesting aspect of this study was the application of coordination polymers for the degradation of selected azo dyes from the binary solution (1:1). It is noteworthy that degradation efficiencies higher than 72%, and over 81% in the presence of H_2_O_2_, respectively, were obtained for both dyes in the binary solution in 240 min, using **CP1**.

The recycling of compound **CP1** for the degradation of AO7 and MO dyes in five cycles of adsorption/degradation demonstrates its efficient reuse. The yield decrease was approximately 30% for AO7 photodegradation and 27% for MO after five reuses, which once again proves the efficiency of **CP1** as a catalyst in the dye degradation process.

In addition, the excellent stability of **CP1** after the photodegradation process of dyes from both simple and binary solutions was demonstrated by the consistency of the initial and final structure of the complex.

A photocatalytic oxidation mechanism was proposed, where *h^+^* and •OH radicals are the main active species in the AO7 and MO photodegradation.

All the obtained results indicate that **CP1** can be used as an environmentally friendly and efficient photocatalyst for the degradation of dyes and consequently for the depollution of wastewater containing dyes.

## Figures and Tables

**Figure 1 molecules-30-01652-f001:**
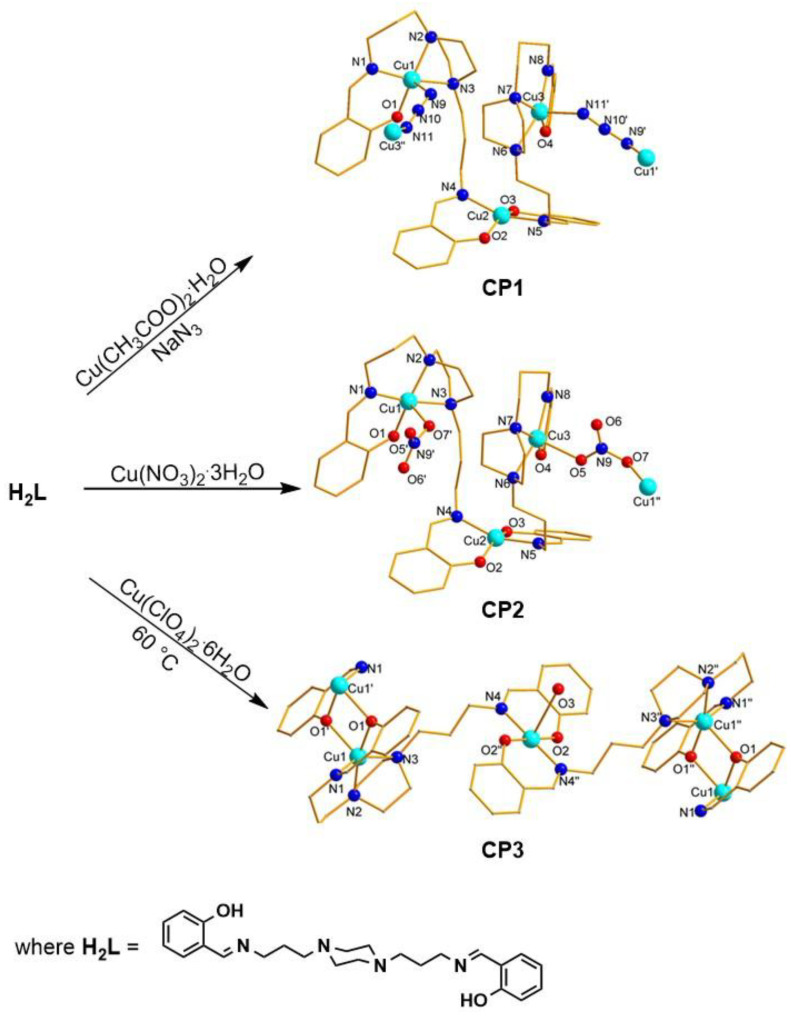
Synthesis of the coordination polymers: **CP1**–**CP3**.

**Figure 2 molecules-30-01652-f002:**
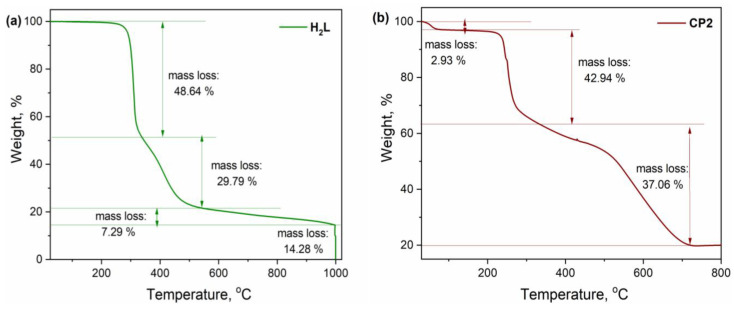
Thermal decomposition of (**a**) **H_2_L** and (**b**) **CP2**.

**Figure 3 molecules-30-01652-f003:**
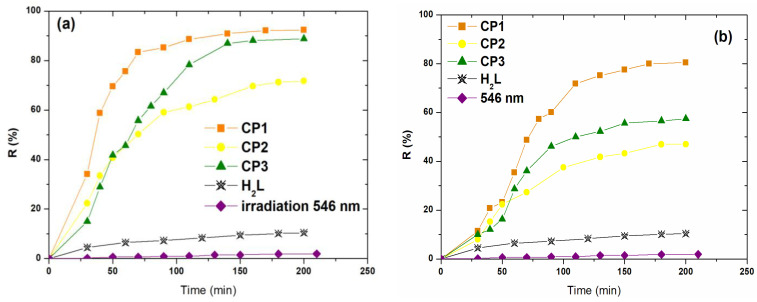
Removal efficiency of **CP1**–**CP3** for the photodegradation of (**a**) Acid Orange 7 and (**b**) Methyl Orange dyes.

**Figure 4 molecules-30-01652-f004:**
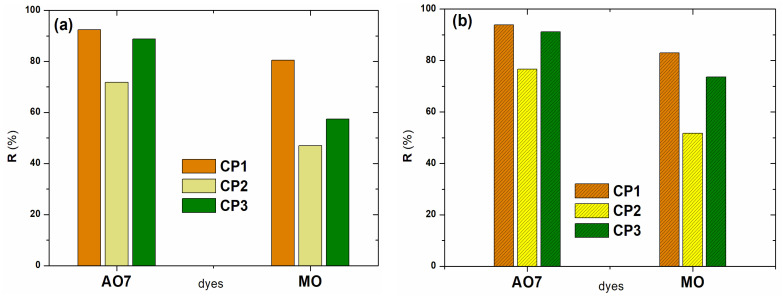
Degradation efficiency of investigated dyes using **CP1**–**CP3**, with (**a**) and without (**b**) adding H_2_O_2_; 1 g/L CPs, 30 mg/L initial dye concentration, 27 °C, solution pH.

**Figure 5 molecules-30-01652-f005:**
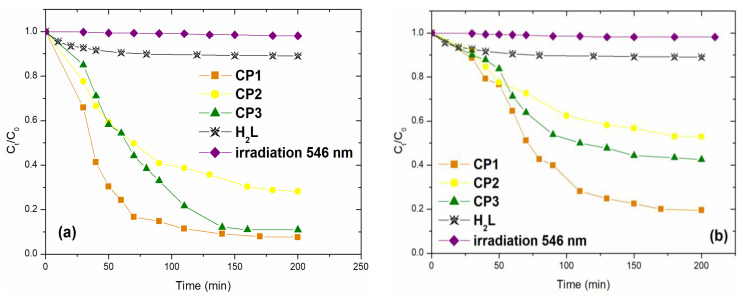
Photodegradation of (**a**) Acid Orange 7 and (**b**) MO dyes in time; under visible light (546 nm), with and without complexes, and in the presence of ligand **H_2_L**.

**Figure 6 molecules-30-01652-f006:**
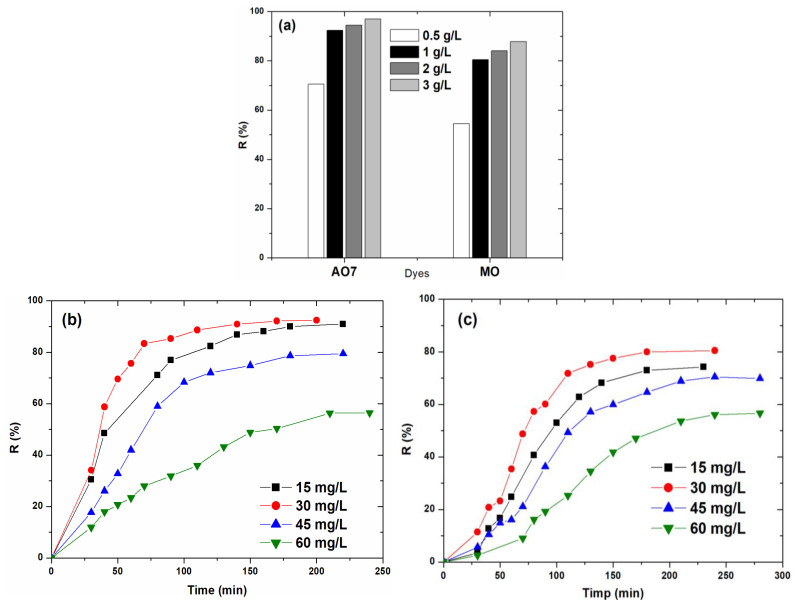
Influence of catalyst quantity (**a**), initial concentration of Acid Orange 7 (**b**), and MO (**c**) dyes on the photodegradation process.

**Figure 7 molecules-30-01652-f007:**
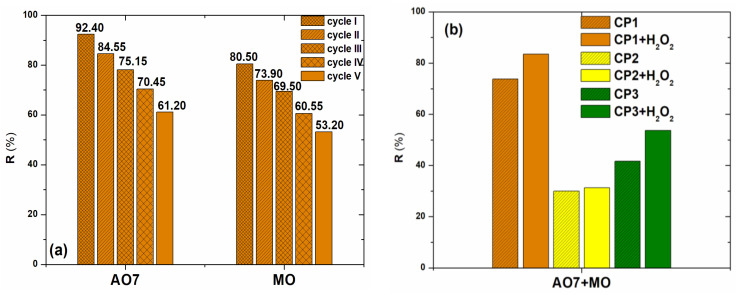
Degradation efficiency (**a**) of investigated dyes in five consecutive cycles; (**b**) of investigated dyes from binary solution using **CP1**–**CP3**, without and with adding H_2_O_2_.

**Figure 8 molecules-30-01652-f008:**
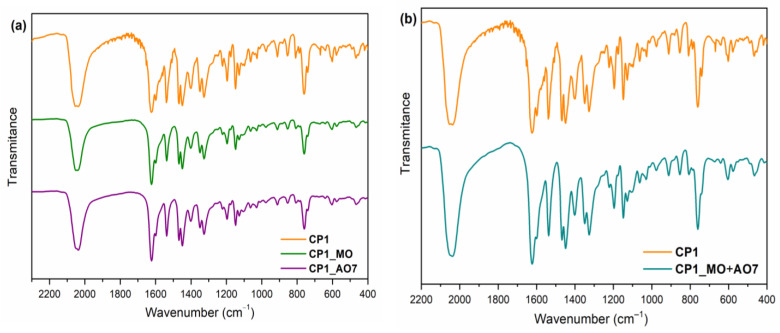
IR spectrum of **CP1** before and after photodegradation of (**a**) single AO7 and MO dyes and (**b**) binary system AO7-MO.

**Figure 9 molecules-30-01652-f009:**
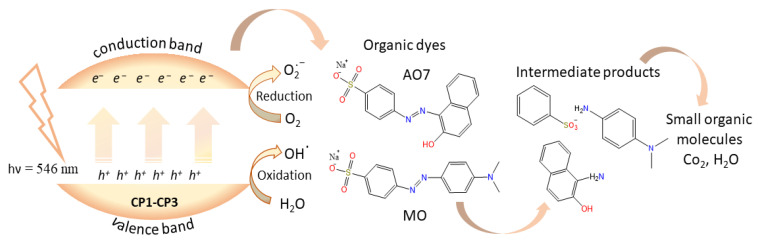
Schematic representation of decomposition mechanism of organic dyes under visible irradiation in the presence of **CP1**–**CP3** and possible intermediate products.

**Table 1 molecules-30-01652-t001:** Photocatalytic efficiency (%) of different coordination polymers for removal of dyes under visible light irradiation.

Dye	Photocatalyst	R (%)	R with H_2_O_2_ (%)	Ref.
Rhodamine B	{[Zn_3_(L)(4,4′-bpy)]}n (GTU-3)	76.50		[31]
Methylene Blue		86.20		
Acid Red 17		52.80		
Crystal Violet	{[Cu_2_(Or)_2_(Bimb)_3_]·4H_2_O}_n_	75.80		[32]
MethylViolet		76.80		
Rhodamine 6G		86.50		
Rose Bengal		76.10		
Methylene Blue		17.80		
Congo Red		40.20		
Methyl blue	[Cd(bpyp)(nba)_2_]	29.24		[33]
Methyl orange		35.44		
Rhodamine B		95.52		
Methyl violet		58.92		
Rhodamine B	{[Cu(L’)Cl]·2.25H_2_O}*n*	6.10	91.20	[47]
Methylene Blue		28.30	91.90	
Methyl orange	{[Cd_3_L_2_(H_2_O)_5_]·H_2_O}n		59.80	[48]
	{[Cd_3_L_2_(hbmb)-(H_2_O)_2_]·2.5H_2_O}n		47.30	
	{[Cd_3_L_2_(btbb)(H_2_O)_2_]·2EtOH·1.5H_2_O}n		51.40	
	{[Cd_6_L_4_(bipy)_2_(H_2_O)_6_]·3H_2_O}n		80.00	
Rhodamine-B	GO@CuO nanocomposite	84.77		[49]
Malachite green		87.81		
Methyl Orange	Fe_3_O_4_@SiO_2_@ZnO		96.00	[50]
Acid Orange 7	**CP-1**	92.40	93.81	This work
	**CP-2**	71.80	76.60	
	**CP-3**	89.03	91.10	
Methyl Orange	**CP-1**	80.50	84.82	This work
	**CP-2**	47.00	51.72	
	**CP-3**	57.45	71.43	

H_6_L_1_ = hexakis(methyl-2-(4-phenoxyphenyl)acetatebenzene)cyclotriphosphazene; 4,4′-bpy = 4,4′-bipyridine; Bimb = 1,4-bis[(1H-imidazol-1-yl)methyl]benzene; OrK = potassium orotate; bpyp = 2,5-bis(pyrid-4-yl)pyridine; Hnba = 4-nitrobenzoic acid; L′ = 4-(2,6-di(pyrazin-2-yl) pyridin-4-yl)benzoate; H_3_L = 3,4-bi(4-carboxyphenyl)-benzoic acid; hbmb = 1,1′-(1,6-hexane) bis(2-methylbenzimidazole); btbb = 1,4-bis(2-(4-thiazolyl)benzimidazole-1-ylmethyl)benzene; 4,4′-bipy = 4,4′-bipyridine).

**Table 2 molecules-30-01652-t002:** Influence of initial concentration on dye removal by photodegradation.

Dye	C_0_ (mg/L)	*R* (%)	Time (min)
AO7	15	90.93	130
	30	92.40	140
	45	79.46	180
	60	56.32	210
MO	15	74.29	170
	30	80.50	180
	45	70.51	220
	60	56.61	240

**Table 3 molecules-30-01652-t003:** Kinetic parameters for AO7 and MO photodegradation by coordination polymers **CP1**–**CP3**, and by **CP1** at different concentrations.

CPs	AO7				MO			
	Conc. (mg/L)	k × 10^3^ (min^−1^)	R^2^	SD	Conc. (mg/L)	k × 10^3^ (min^−1^)	R^2^	SD
**CP1**	15	16.40	0.9716	0.1551	15	11.28	0.9274	0.1483
	30	13.07	0.8631	0.3401	30	9.33	0.9247	0.1707
	45	8.57	0.9263	0.1687	45	6.07	0.9657	0.0906
	60	3.20	0.9918	0.0329	60	3.94	0.9648	0.0581
**CP2**	30	5.25	0.9287	0.1201	30	5.82	0.9220	0.0694
**CP3**	30	9.76	0.9246	0.2319	30	6.70	0.8802	0.1156

**Table 4 molecules-30-01652-t004:** The photocatalytic degradation efficiency of AO7 and MO dyes from binary solution using **CP1**–**CP3** under visible light irradiation.

R (%)										
Dye	AO7				MO				AO7 + MO	
	Single		Binary		Single		Binary			
	-	H_2_O_2_	-	H_2_O_2_	-	H_2_O_2_	-	H_2_O_2_	-	H_2_O_2_
**CP1**	92.40	93.81	74.64	84.05	80.50	84.82	72.05	81.46	73.83	83.54
**CP2**	71.80	76.60	30.12	33.29	47.00	51.72	27.35	29.02	29.98	31.34
**CP3**	89.03	91.10	43.03	55.00	54.45	71.43	34.39	50.51	41.70	53.71

**Table 5 molecules-30-01652-t005:** Characteristics and toxicological information of the dyes.

Dye	Molecular Weight (g/mol)	Structure	Toxicity
AO7	350.32	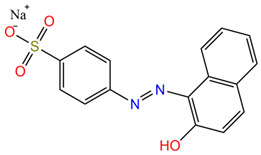	Dangerous for the aquatic environment in the long term. Strong irritant to skin and eyes, and may cause allergic reactions in sensitive people [77,78].
MO	327.33	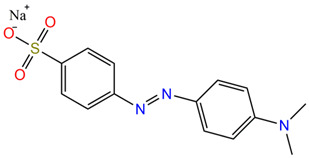	Harmful to the environment, biology, and human health. May cause respiratory tract irritation and skin irritation [79]. It is a toxic, carcinogenic, tumorigenic, mutagenic, and genotoxic azo dye [80].

## Data Availability

The authors declare that the data supporting the findings of this study are available within the paper and its Appendix A. CCDC 2006145 (**CP1**), 2006144 (**CP2**), and 2006146 (**CP3**) contain the supplementary crystallographic data for this paper [43]. These data can be obtained free of charge via http://www.ccdc.cam.ac.uk/structures/ (accessed on 1 April 2025) (or from the Cambridge Crystallographic Data Centre, 12 Union Road, Cambridge CB2 1EZ, UK; fax: (+44)1223-336-033; or deposit@ccdc.cam.uk).

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
