# Peer review of "Efficient Photodegradation of Dyes from Single and Binary Aqueous Solutions Using Copper(II) Coordination Polymers"

_molecules, 2025, doi:10.3390/molecules30081652_

Round 1
Reviewer 1 Report
Comments and Suggestions for Authors
Dye wastewater and organic pollutants remediation is a global challenge. This work offers a novel strategy by designing reusable photocatalysts, particularly for the remediation of real-world industrial effluents with mixed pollutants. The application of copper(II) coordination polymers (CP1-CP3) as photocatalysts for the photocatalytic degradation of both single and binary dye systems (Acid Orange 7 and Methyl Orange) under visible light has a certain degree of innovation and significance. While, there are some issues required to be revised. I recommend this manuscript for publish after a minor revision of the following main issues and concerns:
- The first appearance of CP1 and CP2should be accompanied with their full standard chemical formula, not just abbreviation of “Me”, and should give the full chemical structure of L, not just H2L.
- The resolution of almost all the scheme and figures are low and hard to get the details.
- Horizontal arrangement of Figure 1a and Figure 1b may be better.
- Thermal decomposition of CP1 is missing, is the result of the decomposition of CP1 has any different from that of CP2. The decomposition result of CP1 can tell its different stability from CP2.
- Figures 2-4. Normally, figure needs to be enclosed with up, down, left, and right coordinate axes. If other pictures are enclosed by four coordinate axes, all the figures in the manuscript should be in a unified format.
- The Schematic representation of decomposition mechanism of organic dyes under visible irradiation in the presence of CP1-CP3 is too simple. What are the intermediate products?
- Degradation efficiency suffered an obvious decrease after five consecutive cycles (Figure 6). Is there any possible that the ligands of CP1-CP3 were also oxidated by ROS, thus leading to a decrease in degradation efficiency after several recycles? This can be verified by monitoring the concentration of free copper ions or fragments of CP1-CP3 in the solution.
- Grammatical issues, “The band gaps for CP1-CP3 was..”, revised into “The band gaps for CP1-CP3 were calculated...”.
Author Response
|
Response to Reviewers Comments |
|
Reviewer 1 Dye wastewater and organic pollutants remediation is a global challenge. This work offers a novel strategy by designing reusable photocatalysts, particularly for the remediation of real-world industrial effluents with mixed pollutants. The application of copper(II) coordination polymers (CP1-CP3) as photocatalysts for the photocatalytic degradation of both single and binary dye systems (Acid Orange 7 and Methyl Orange) under visible light has a certain degree of innovation and significance. While, there are some issues required to be revised. I recommend this manuscript for publish after a minor revision of the following main issues and concerns: Response to Comments for Reviewer 1 |
|
Thank you very much for taking the time to review this manuscript. Please find the detailed responses below and the corresponding revisions/corrections highlighted/in track changes in the re-submitted files.
|
|
1. Comments 1: The first appearance of CP1 and CP2 should be accompanied with their full standard chemical formula, not just abbreviation of “Me”, and should give the full chemical structure of L, not just H2L. |
|
Response 1: Thank you for pointing this out. The full chemical formulas for compounds CP1-CP3 have been corrected, both in the abstract and introduction sections. The chemical structure of L is identical to that of H2L, except for the absence of two hydrogen atoms, resulting in L2-. Bellow you can see the modified version of Figure 1 with the chemical structure of the ligand in the form L2-:
From the perspective of synthetic chemistry, we consider the representation of H2L to be the correct one from a chemical point of view. If you consider it essential and the editor considers it appropriate, we can modify the chemical structure of the ligand in the revised form of our paper.
|
|
2. Comments 2: The resolution of almost all the scheme and figures are low and hard to get the details. |
|
Response 2: Thank you for your opinion. Accordingly, we have modified the quality/resolution of all figures (600 dpi) to better represent the data obtained and the details.
|
|
Comments 3: Horizontal arrangement of Figure 1a and Figure 1b may be better |
|
Response 3: Thank you for suggestion. In the revised form, the horizontal arrangement of Figures 1a and 1b was displayed.
|
|
Comments 4: Thermal decomposition of CP1 is missing, is the result of the decomposition of CP1 has any different from that of CP2. The decomposition result of CP1 can tell its different stability from CP2. |
|
Response 4: As we already mentioned in the 2.1.1 section “Due to the fact that the perchlorate and azide metal complexes are potentially explosive, the thermal properties were investigated only for H2L ligand and CP2 [52].” due to the presence of the azide anion as a co-ligand in the structure of CP1, and considering that azide metal complexes can be potentially explosive, thermal decomposition of this compound could not be performed.
|
|
Comments 5: Figures 2-4. Normally, figure needs to be enclosed with up, down, left, and right coordinate axes. If other pictures are enclosed by four coordinate axes, all the figures in the manuscript should be in a unified format. |
|
Response 5: At the reviewer's recommendation, all figures were modified and presented in the revised paper in a unified format.
|
|
3. Comments 6: The Schematic representation of decomposition mechanism of organic dyes under visible irradiation in the presence of CP1-CP3 is too simple. What are the intermediate products? |
|
Response 6: The schematic representation of the decomposition mechanism of organic dyes under visible irradiation in the presence of CP1-CP3 has been modified and presented in the revised paper. AO7 and MO are very stable compounds under light irradiation when no catalyst is available (as seen in Figure 2). The reactive oxygen species (ROS) generated were strong oxidants that could oxidize the AO7 and MO dyes. The degradation mechanism possibly involved the symmetric cleavage of the azo bond (–N=N–) producing: β-naphthol, N,N-dimethyl-p-phenylenediamine and sulfanilic acid [https://doi.org/10.1038/srep41963; https://doi.org/10.3390/catal11040428], followed by further degradation of the intermediates products into: carbon dioxide (CO2), water (H2O) and other degradation products.
|
|
Comments 7: Degradation efficiency suffered an obvious decrease after five consecutive cycles (Figure 6). Is there any possible that the ligands of CP1-CP3 were also oxidated by ROS, thus leading to a decrease in degradation efficiency after several recycles? This can be verified by monitoring the concentration of free copper ions or fragments of CP1-CP3 in the solution. |
|
Response 7: Thank you for pointing this out. Yes, it is possible for the reactive oxygen species (ROS) interact with the imine bond from the Schiff base ligand. The oxidation of the imine group can disrupt the ligand’s ability to coordinate with the metal center, leading to alteration of the polymer structure. In some cases, the ROS might also interact with the metal center, leading to changes in the coordination environment, and possible releasing metal ions. To accurately monitor the concentration of free copper ions, a combination of techniques may be necessary to obtain reliable data, including UV-Vis spectroscopy, Atomic Absorption Spectroscopy, and Ion-Selective Electrode measurements. However, we currently only have access to UV-Vis spectroscopy, where Cu²⁺ ions exhibit strong absorption in the UV region. Unfortunately, the presence of intense absorption bands from the Schiff base ligands in the same region complicates the detection of copper ion absorption bands. Due to the lack of adequate equipment and the time frame given for the review, we are unable to investigate this issue further.
|
|
4. Comments 8: Grammatical issues, “The band gaps for CP1-CP3 was..”, revised into “The band gaps for CP1-CP3 were calculated...”. |
|
Response 8: Thank you for the observation. We checked and corrected in the revised manuscript |
Considering all the modifications we made, we hope that this version of the manuscript is now suitable to be published in Molecules journal.

Reviewer 2 Report
Comments and Suggestions for Authors
This article is a study on the application of three copper(II) coordination polymers as catalysts for photocatalytic degradation of two types of orange dyes, from both single and binary aqueous solutions. Of the three polymers considered, the one labeled as CP1 shows the highest photodegradation efficiency for both types of dyes: Acid Orange 7 and Methyl Orange.
The authors also propose a photocatalytic oxidation mechanism, while also studying the stability of the CP! complex, for both solutions and before and after the process. The paper is well structured and uses clear and direct language.
The methods section is well organized and allows the reader to follow the processes described in a clear way. The Conclusions follow logically from the results and make no inferences that can't be traced back to them. The topic is of interest and relevance to the environmental problems that Earth and life on it are facing right now.
Minor revisions:
- The figures labelled as "scheme" should simply be labeled as figures, like all the rest.
- The notation needs to be uniformized throughout the manuscript. For instance, in Section 2.1.1, page 5, equation (1) introduces the term αhν, which then in line 132 is presented as αhv, with a v instead of the Greek letter nu (ν). Also the quadratic power in that term should be properly written.
- This may be just a matter of graphic design that can be fixed later, but in general, all figures are presented pixelated and in a rather low resolution that makes them hard to read. This is particularly evident in Figure 2.
- The manuscript should be proof read for typos and grammar mistakes, e.g., In page 10, line 256, there is a repeated word: "As can be seen in in Fig. 5b..."
All these are quite minor revisions, I consider the paper to be well written and interesting overall.
Author Response
|
Response to Reviewers Comments |
|
Reviewer 2 This article is a study on the application of three copper(II) coordination polymers as catalysts for photocatalytic degradation of two types of orange dyes, from both single and binary aqueous solutions. Of the three polymers considered, the one labeled as CP1 shows the highest photodegradation efficiency for both types of dyes: Acid Orange 7 and Methyl Orange. The authors also propose a photocatalytic oxidation mechanism, while also studying the stability of the CP! complex, for both solutions and before and after the process. The paper is well structured and uses clear and direct language. The methods section is well organized and allows the reader to follow the processes described in a clear way. The Conclusions follow logically from the results and make no inferences that can't be traced back to them. The topic is of interest and relevance to the environmental problems that Earth and life on it are facing right now. |
|
Response to Reviewer 2 Comments |
|
Thank you very much for taking the time to review this manuscript. Please find the detailed responses below and the corresponding revisions/corrections highlighted/in track changes in the re-submitted files.
|
|
Comments 1: The figures labelled as "scheme" should simply be labeled as figures, like all the rest. |
|
Response 1: Thank you for the suggestion. Therefore, in the revised paper, we have made the suggested changes, renamed "scheme 1" and "scheme 2" and renumbered the figures accordingly. |
|
Comments 2: The notation needs to be uniformized throughout the manuscript. For instance, in Section 2.1.1, page 5, equation (1) introduces the term αhν, which then in line 132 is presented as αhv, with a v instead of the Greek letter nu (ν). Also the quadratic power in that term should be properly written. |
|
Response 2: Thank you for the observation. We checked and corrected in the revised manuscript. |
|
Comments 3: This may be just a matter of graphic design that can be fixed later, but in general, all figures are presented pixelated and in a rather low resolution that makes them hard to read. This is particularly evident in Figure 2. |
|
Response 3: Thank you for the advice. Accordingly, we have modified the quality/resolution of all figures (600 dpi) to better represent the data obtained and the details. |
|
Comments 4: The manuscript should be proof read for typos and grammar mistakes, e.g., In page 10, line 256, there is a repeated word: "As can be seen in in Fig. 5b..." |
|
Response 4: Thank you for the observation. We checked and corrected in the revised manuscript |
|
|
|
All these are quite minor revisions, I consider the paper to be well written and interesting overall. |
|
|
Considering all the modifications we made, we hope that this version of the manuscript is now suitable to be published in Molecules journal.

Reviewer 3 Report
Comments and Suggestions for Authors
The synthesis and structures of the three coordination polymers mentioned in this manuscript have already been published in Polyhedron 190 (2020) 114766. This manuscript is thus not suitable for publication in Molecules, although the photodegradation may be interesting.
Author Response
|
Response to Reviewers Comments |
|
Reviewer 3 |
|
The synthesis and structures of the three coordination polymers mentioned in this manuscript have already been published in Polyhedron 190 (2020) 114766. This manuscript is thus not suitable for publication in Molecules, although the photodegradation may be interesting. |
|
Response to Reviewer 3 Comments |
|
We appreciate your opinion. As we already mentioned in submitted paper: “The detailed information about the synthesis and structural characterization of CP1-CP3 are presented in the previous paper [43]”. We hope that this will not be a decisive factor in making the final decision. Our article is “a continuation of our work [43]” in which the possible application of the obtained coordination polymers as catalyst was investigated and demonstrated. Since our paper was submitted to the special issue “Study on Synthesis and Photochemistry of Dyes” which belongs to the section "Photochemistry" we believe that our work is suitable for publication in Molecules, is stimulating and provocative, and deserving of a wide readership based on the following main points: (1) It is the first time that coordination polymers are investigated for the degradation of dyes in binary systems. (2) The high degradation efficiency of 72 % and over 81% (H2O2) respectively obtained for the both dyes in the binary solution. (3) The excellent stability of CP1 after the photodegradation process was demonstrated by the consistency of the initial and final structure of the complex. (4) The recycling of compound CP1 for the degradation of investigated dyes in five cycles of adsorption/degradation demonstrates its efficient reuse. Our paper presents experimental and practical research connecting environmental sciences with materials science. |

Round 2
Reviewer 1 Report
Comments and Suggestions for Authors
The revised manuscript is recommended for publish.
Reviewer 2 Report
Comments and Suggestions for Authors
The authors have addressed all the comments from my previous review. I consider this manuscript can be published as is. Thank you.
Reviewer 3 Report
Comments and Suggestions for Authors
I would be happy to change my recommendation to "accept" if the editors think this manuscript fits into the scope of the journal.